# Robot Reinforcement Learning on the Constraint Manifold

**Puze Liu**[1] , **Davide Tateo**[1], **Haitham Bou-Ammar**[2] **and Jan Peters**[1]
[1] Department of Computer Science, Technische Universität Darmstadt, Germany
[2] Huawei R&D London, United Kingdom
{puze, davide}@robot-learning.de,
haitham.ammar@huawei.com, jan.peters@tu-darmstadt.de

**Abstract:** Reinforcement learning in robotics is extremely challenging due to many practical issues, including safety, mechanical constraints, and wear and tear. Typically, these issues are not considered in the machine learning literature. One crucial problem in applying reinforcement learning in the real world is Safe Exploration, which requires physical and safety constraints satisfaction throughout the learning process. To explore in such a safety-critical environment, leveraging known information such as robot models and constraints is beneficial to provide more robust safety guarantees. Exploiting this knowledge, we propose a novel method to learn robotics tasks in simulation efficiently while satisfying the constraints during the learning process.

**Keywords:** Robot Learning, Constrained Reinforcement Learning, Safe Exploration

## 1 Introduction

Despite the notable success of Deep Reinforcement Learning (RL) in solving complex tasks in the discrete world, video games, as well as continuous control problems in simulation [1, 2, 3, 4], applying RL in the real world remains a challenging task. One important factor that cannot be neglected in real-world applications is the necessity of satisfying constraints. Many practical considerations can be formulated in the form of constraints, such as safety and mechanical viability. For example, in the robot manipulation task, the robot should not take actions that damage the environment and can not take actions that exceed its feasible range. However, typical RL algorithms, which maximize the cumulative reward by continuous trial and error, do not take into account the satisfaction of constraints during the exploration process. Exploring the environment while meeting the constraints is a challenging problem.

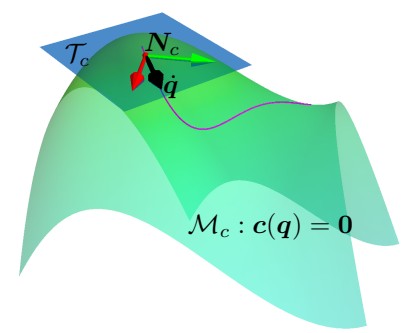

Figure 1: Acting on the Tangent Space of the Constraint Manifold. The constraint set $c(q) = 0$ is a differentiable manifold $\mathcal{M}_c$ embedded in the original state space. We use a set of basis vectors $N_c$ to represent the span of tangent space $\mathcal{T}_c$. The tangent space velocity/acceleration can be determined by a coordinate based on the bases, and the control action is determined based on the tangent space velocity/acceleration, the resulting trajectory is maintained on the constraint manifold.

Safe exploration is an significant field of RL which requires to comply with the constraints during the whole learning process [5]. There are several safe exploration frameworks in the literature: a possible direction is proposed in [6, 7] that relies on prior knowledge (policies, value functions) to initialize the system in a safe region and gradually increase the area of exploration using new information obtained from the environment. Other approaches rely on the definition of a safe policy [8, 9], which tries to pull the agent back to a safe state. However, these choices require excessive work in defining such policy, and safe policies could conflict against each other when multiple constraints are violated. Finally, other methods incorporate model information of constraints with model-free RL algorithms and do not require

5th Conference on Robot Learning (CoRL 2021), London, UK.

the definition of a manual policy [10, 11, 12]. In these approaches, the agent tries to find the feasible action using constrained optimization techniques at each time step.

In this paper, we propose a novel method, Acting on the TAngent Space of the Constraint Manifold (ATACOM), which the agent explores in the tangent space of the constraint manifold, as shown in Figure 1. The proposed method convert the constrained RL problem to a typical unconstrained RL problem. This method allows us to utilize *any* model-free RL algorithms while maintaining the constraints below the tolerance. Furthermore, ATACOM can handle both equality and inequality constraints. For example, in the task of a robot wiping a table, the end-effector should move on the surface of the table (equality constraints) while the joint positions and velocities are within its joint limits (inequality constraints). In addition, for tasks with equality constraints, our method explores the *lower-dimensional* manifold embedded in the original action space. To test our method, we demonstrate three different tasks, *CircleMoving*, *PlanarAirHockey*, and *IiwaAirHockey*[13], with different combinations of equality and inequality constraints. We test five state-of-the-art model-free RL algorithms (PPO, TRPO, DDPG, TD3, SAC) in each environment. The result shows that all algorithms can learn the policy efficiently while maintaining the constraints below the tolerance.

The advantage of ATACOM can be summarized as follows: (i) can deal both with **equality and inequality constraints**. All of the constraints at each time step are maintained below the tolerance during the whole learning process. (ii) does not require an initial feasible policy, the agent can **learn from scratch**. (iii) requires **no manual safe backup policy** to move the system back into the safe region. (iv) can be applied to any model-free RL algorithm, using both **deterministic and stochastic policies**. (v) can focus the exploration on the **lower-dimensional manifold** instead of exploring in the original action space for equality constrained problem. (vi) have **better learning performance** as the inequality constraints restrict to a smaller feasible state-action space. As a downside, our method requires: (i) differentiable constraint functions. (ii) a sufficient accurate invertible dynamics model of the robot or a well-performed tracking controller. Videos and code can be found: https://sites.google.com/view/robot-air-hockey/atacom

**Related Work.** In the last decades, Constrained Markov Decision Processes (CMDP) [14] has attracted a lot of interest from RL researchers, to solve constrained control problems. Under this framework, several different forms of constraints have been studied. One important form of constraint is the expected cost below a threshold. Many works maximize the expected return while maintaining the expected cost below a threshold [11, 15, 16, 17, 18, 19, 20, 21]. Different types of constrained optimization techniques are applied in the policy update process. Achiam et al. proposed a trust-region method Constrained Policy Optimization (CPO) inspired from Trust Region Policy Optimization (TRPO) [19]. Liu et al. proposed the interior point method for policy optimization [16]. Another type of approach is to adapt the Lagrangian relaxation method for the constrained RL setting [14, 17, 15, 18]. Lastly, Chow et al. proposed a method to generate the Lyapunov function that guarantees constraints satisfaction [11, 22]. These approaches focus on the constraint of the cumulative cost and require an initial feasible policy. However, this cumulative cost criterion cannot ensure safety for tasks where avoiding catastrophic failures is crucial, e.g., car crashing.

Other approaches focus on the state dependant constraints, which should be fulfilled at every time step. To meet this requirement, safe exploration methods can be employed. Garcia, et al., proposed a method based on a risk function and a baseline agent, where the control action is sampled based on the evaluation of the risk [6]. The shielding [8] and backup policy [9] frameworks interfere with the control action to pull the system back to the safe states. These approaches require a manual defined safe policy. Berkenkamp, et al. [7], Wachi, et al. [23], Koller, et al. [24], and Hewing, et al.[25] proposed model-based approaches to ensure the safety. These approaches start from an initial feasible policy and progressively increase the safe region based on the learned dynamics model. Recent methods also try to incorporate the model and the constraint information with the model-free RL algorithms. Dalal, et al., added a safe layer which analytically finds the closest action w.r.t the policy derived one [10]. Cheng, et al., proposed a barrier function method to guarantee safety during the exploration [12]. Finally, other approaches has also address the safety issue from different perspectives, such as the policy composition [26, 27] and reachability-based approach [28, 29, 30].

Our approach considers the second group of constraints. However, different from other comparable methods, ATACOM does not require an initial policy, it can learn from scratch. In addition, our method does not require a backup policy either, as the constraint violations are forecasted and corrected at each step. Furthermore, our method is not specifically restricted to any learning algorithm.

## 2 Learning on the Constraint Manifold

In this section, we discuss ATACOM in detail. We first introduce the mathematical notation used in this paper. Then, to demonstrate the core concept, we start with a simple scenario that the constraint on the subset of the state variable $q$ and the action can be formulated as a function of the state velocity $a = \Lambda(\dot{q})$. Next, considering the continuity of velocity (sampling over the velocity does not ensure the continuity), we convert the original state constraint to a viability constraint that incorporates the velocity of the constraint. The action is chosen as a function of the acceleration $a = \Lambda(\ddot{q})$. From a robotics point of view, this $a$ can be the torque applied to each joint, and $\Lambda$ is the inverse dynamics model. Then, to cope with the velocity limit, we add the viability condition to the acceleration. Lastly, we discuss some important practical issues of ATACOM, such as the error correction, the tangent space convention that determines the null space bases.

**Definitions** We consider the CMDP with continuous state-action space. A CMDP is a tuple $(\mathcal{S}, \mathcal{A}, P, R, \gamma, \mathcal{C})$, where $\mathcal{S}$ is a state space, $\mathcal{A}$ is an action space, $P : \mathcal{S} \times \mathcal{A} \times \mathcal{S} \rightarrow [0, 1]$ is a transition kernel, $\gamma$ is a discount factor, and $\mathcal{C} : \{c_i : \mathcal{S} \rightarrow \mathbb{R} | i \in 1, ..., k\}$ is a set of *immediate state-constraint* functions.

**Assumption** In this paper, we decompose the state variable $s \in \mathcal{S}$ into the directly controllable state $q \in \mathcal{Q}$ and uncontrollable state $x \in \mathcal{X}$, i.e., $s = [q\ x]^\mathsf{T}$. We assume that the constraints $c(q) \leq 0$ are known and depend purely on the controllable state. In addition, we assume that the action $a$ can be determined based on the $i$-th order time derivative of the controllable state, i.e., $a = \Lambda(q^{(i)}), i \in \{1, 2, ...\}$. For example, we can determine the joint torque using an inverse dynamics model or send the desired positions/velocities obtained via integration to a tracking controller (e.g., PID controller). The general form of the constrained reinforcement learning problem can be formulated as

$$\max_{\theta} \mathbb{E}_{s_t, a_t} \left[ \sum_{t=0}^{T} \gamma^t r(s_t, a_t) \right], \qquad \text{s.t.} \quad c(q_t) \leq 0.$$

### 2.1 State Constraints

The state constraints are defined as

$$f(q) = 0, \quad g(q) \leq 0, \tag{1}$$

where $f : \mathbb{R}^Q \rightarrow \mathbb{R}^F, g : \mathbb{R}^Q \rightarrow \mathbb{R}^G$ are two $C^2$ mappings for $F$ equality and $G$ inequality constraints, and $F < Q$. We add the slack variables $\mu \in \mathbb{R}^G$ in inequality constraints to convert the original constraints (1) into equality constraints

$$c(q, \mu) = \begin{bmatrix} f(q) & g(q) + \frac{1}{2}\mu^2 \end{bmatrix}^\mathsf{T} = 0. \tag{2}$$

The constraint set (2) is a $(F + G)$ dimensional manifold embedded in $(Q + G)$ dimensional space. We calculate the time derivative of (2)

$$\dot{c}(q, \mu, \dot{q}, \dot{\mu}) = \begin{bmatrix} J_f(q) & 0 \\ J_g(q) & \text{diag}(\mu) \end{bmatrix} \begin{bmatrix} \dot{q} \\ \dot{\mu} \end{bmatrix} = J_c(q, \mu) \begin{bmatrix} \dot{q} \\ \dot{\mu} \end{bmatrix}, \tag{3}$$

with the Jacobians $J_f \in \mathbb{R}^{F \times Q}$ and $J_g \in \mathbb{R}^{G \times Q}$ of $f(q)$ and $g(q)$, respectively. Both Jacobians are combined into the Jacobian Matrix $J_c(q, \mu) \in \mathbb{R}^{(F+G) \times (Q+G)}$ of the complete constraint set.

We can find the null space matrix $N_c(q, \mu) = \text{Null}[J_c(q, \mu)] \in \mathbb{R}^{(Q+G) \times (Q-F)}$ via SVD [31] or QR [32] decomposition, such that $J_c(q, \mu)N_c(q, \mu) = 0$. Each column of the orthogonal matrix $N_c(q, \mu)$ represents a basis vector of the null space of $J_c(q, \mu)$. These null space bases can also be viewed as the tangent space bases of the constraint manifold as illustrated in Figure 1. We can construct a tangent space velocity of the constraint manifold by a coordinate $\alpha$ as

$$\begin{bmatrix} \dot{q}_\mathcal{T} \\ \dot{\mu}_\mathcal{T} \end{bmatrix} = N_c(q, \mu)\alpha, \tag{4}$$

Substituting $[\dot{q}\ \dot{\mu}]^\mathsf{T}$ of (3) by $[\dot{q}_\mathcal{T}\ \dot{\mu}_\mathcal{T}]^\mathsf{T}$ of (4), we have the constraint velocity

$$\dot{c}(q, \mu, \dot{q}, \dot{\mu}) = J_c(q, \mu)N_c(q, \mu)\alpha = 0. \tag{5}$$

Equation (5) implies that the constraints do not change regardless of the choice of $\alpha$. Based on this concept, the ATACOM method can be summarized as follows: Starting from a feasible point

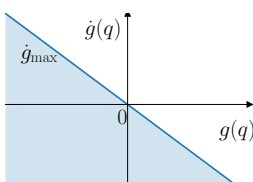

Figure 2: Viability Constraints

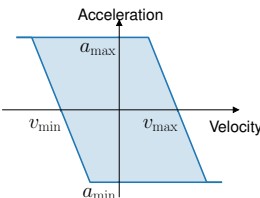

Figure 3: Feasible Acceleration Region

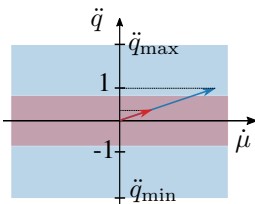

Figure 4: Tangent space bases

$(\boldsymbol{q}(0), \boldsymbol{\mu}(0)) \in \{(\boldsymbol{q}, \boldsymbol{\mu}) | \boldsymbol{c}(\boldsymbol{q}, \boldsymbol{\mu}) = \boldsymbol{0}\}$, we choose the tangent space velocity $[\dot{\boldsymbol{q}}_\mathcal{T}(t), \dot{\boldsymbol{\mu}}_\mathcal{T}(t)]^\mathsf{T} = \boldsymbol{N}_c(\boldsymbol{q}(t), \boldsymbol{\mu}(t))\boldsymbol{\alpha}(t)$ and the corresponding action as $\boldsymbol{a}(t) = \boldsymbol{\Lambda}(\dot{\boldsymbol{q}}_\mathcal{T}(t))$. Thus, the constrained RL problem is converted into an unconstrained RL problem. The resulting trajectory $\boldsymbol{q}(t)$ satisfies the constraints $\boldsymbol{c}(\boldsymbol{q}(t), \boldsymbol{\mu}(t)) = \boldsymbol{0}$.

## 2.2 Viability Constraints

For a physical system, it is often required a continuous velocity command. However, directly sampling velocities $\dot{\boldsymbol{q}}$ does not ensure this continuity. A simple solution is to sample accelerations, apply force to the system or determine the velocity via integration. Furthermore, when considering inequality constraints, it is also desirable that $\dot{\boldsymbol{g}}(\boldsymbol{q}, \dot{\boldsymbol{q}}) \leq \boldsymbol{0}$ when $\boldsymbol{g}(\boldsymbol{q}) = \boldsymbol{0}$ to avoid overshooting. We convert the original state constraints (1) to *viability constraints* inspired by the linear viability condition [33]

$$\boldsymbol{f}(\boldsymbol{q}) + \boldsymbol{K}_f \dot{\boldsymbol{f}}(\boldsymbol{q}, \dot{\boldsymbol{q}}) = \boldsymbol{f}(\boldsymbol{q}) + \boldsymbol{K}_f \boldsymbol{J}_f(\boldsymbol{q})\dot{\boldsymbol{q}} = \boldsymbol{0},$$
$$\boldsymbol{g}(\boldsymbol{q}) + \boldsymbol{K}_g \dot{\boldsymbol{g}}(\boldsymbol{q}, \dot{\boldsymbol{q}}) = \boldsymbol{g}(\boldsymbol{q}) + \boldsymbol{K}_g \boldsymbol{J}_g(\boldsymbol{q})\dot{\boldsymbol{q}} \leq \boldsymbol{0}, \tag{6}$$

with diagonal matrices $\boldsymbol{K}_f \in \mathbb{R}^{F \times F}, \boldsymbol{K}_g \in \mathbb{R}^{G \times G}$ having all positive entries. The matrices $\boldsymbol{K}_f$ and $\boldsymbol{K}_g$ determine the maximum velocities of the constraints $\dot{\boldsymbol{f}}$ and $\dot{\boldsymbol{g}}$ w.r.t to the value of the constraints. The viability constraint of the inequality constraint is illustrated in Figure 2. When $g(q) < 0$, the upper bound of the constraint velocity is $\dot{g}_{\max} > 0$ which means that it is still possible to get close to the constraint boundary. However, if $g(q) > 0$, the upper bound of constraint velocity $\dot{g}_{\max}$ should be smaller than zero to pull the violations back.

Analogous to the derivations from equation (2) and (3), we have

$$\boldsymbol{c}(\boldsymbol{q}, \dot{\boldsymbol{q}}, \boldsymbol{\mu}) = \begin{bmatrix} \boldsymbol{f}(\boldsymbol{q}) + \boldsymbol{K}_f \boldsymbol{J}_f(\boldsymbol{q})\dot{\boldsymbol{q}} \\ \boldsymbol{g}(\boldsymbol{q}) + \boldsymbol{K}_g \boldsymbol{J}_g(\boldsymbol{q})\dot{\boldsymbol{q}} + \frac{1}{2}\boldsymbol{\mu}^2 \end{bmatrix} = \boldsymbol{0}, \tag{7}$$

and

$$\dot{\boldsymbol{c}}(\boldsymbol{q}, \dot{\boldsymbol{q}}, \ddot{\boldsymbol{q}}, \boldsymbol{\mu}, \dot{\boldsymbol{\mu}}) = \underbrace{\begin{bmatrix} \boldsymbol{K}_f \boldsymbol{J}_f(\boldsymbol{q}) & \boldsymbol{0} \\ \boldsymbol{K}_g \boldsymbol{J}_g(\boldsymbol{q}) & \mathrm{diag}(\boldsymbol{\mu}) \end{bmatrix}}_{\boldsymbol{J}_c(\boldsymbol{q}, \boldsymbol{\mu})} \begin{bmatrix} \ddot{\boldsymbol{q}} \\ \dot{\boldsymbol{\mu}} \end{bmatrix} + \underbrace{\begin{bmatrix} \boldsymbol{J}_f(\boldsymbol{q})\dot{\boldsymbol{q}} + \boldsymbol{K}_f \boldsymbol{b}_f(\boldsymbol{q}, \dot{\boldsymbol{q}}) \\ \boldsymbol{J}_g(\boldsymbol{q})\dot{\boldsymbol{q}} + \boldsymbol{K}_g \boldsymbol{b}_g(\boldsymbol{q}, \dot{\boldsymbol{q}}) \end{bmatrix}}_{\boldsymbol{\psi}(\boldsymbol{q}, \dot{\boldsymbol{q}})} = \boldsymbol{0}, \tag{8}$$

where $\boldsymbol{b}_f(\boldsymbol{q}, \dot{\boldsymbol{q}}) = \dot{\boldsymbol{q}}^\mathsf{T} \boldsymbol{H}_f(\boldsymbol{q})\dot{\boldsymbol{q}}$, $\boldsymbol{b}_g(\boldsymbol{q}, \dot{\boldsymbol{q}}) = \dot{\boldsymbol{q}}^\mathsf{T} \boldsymbol{H}_g(\boldsymbol{q})\dot{\boldsymbol{q}}$ and $\boldsymbol{H}_f \in \mathbb{R}^{F \times Q \times Q}, \boldsymbol{H}_g(\boldsymbol{q}) \in \mathbb{R}^{G \times Q \times Q}$ are Hessians of $\boldsymbol{f}(\boldsymbol{q}), \boldsymbol{g}(\boldsymbol{q})$, respectively. We can construct the joint acceleration as

$$\begin{bmatrix} \ddot{\boldsymbol{q}} \\ \dot{\boldsymbol{\mu}} \end{bmatrix} = -\boldsymbol{J}_c^\dagger(\boldsymbol{q}, \boldsymbol{\mu})\boldsymbol{\psi}(\boldsymbol{q}, \dot{\boldsymbol{q}}) + \boldsymbol{N}_c(\boldsymbol{q}, \boldsymbol{\mu})\boldsymbol{\alpha}, \tag{9}$$

with the pseudo-inverse $\boldsymbol{J}_c^\dagger(\boldsymbol{q}, \boldsymbol{\mu})$ and the null space matrix $\boldsymbol{N}_c(\boldsymbol{q}, \boldsymbol{\mu})$ of the Jacobian $\boldsymbol{J}_c(\boldsymbol{q}, \boldsymbol{\mu})$, respectively. The first term in equation (9) is the necessary acceleration that maintains the curvature of the constraints manifold (7) and the second term is the tangent space acceleration of the constraints. When starting from the point $[\boldsymbol{q}(0), \dot{\boldsymbol{q}}(0), \boldsymbol{\mu}(0)] \in \{(\boldsymbol{q}, \dot{\boldsymbol{q}}, \boldsymbol{\mu}) | \boldsymbol{c}(\boldsymbol{q}, \dot{\boldsymbol{q}}, \boldsymbol{\mu}) = \boldsymbol{0}\}$ and sampling over $\boldsymbol{\alpha}$, the joint acceleration $\ddot{\boldsymbol{q}}$ and the corresponding action $\boldsymbol{a}$ satisfy the constraints.

## 2.3 Viability Acceleration Bound

In robotics as well as other mechanical systems, it is important to consider the velocity constraints of the actuator. Also, the acceleration should be bounded properly to avoid overshooting. We again use the concept of viability to determine the upper and lower bound of the acceleration

$$\boldsymbol{a}_u = \max\left(\min\left(\boldsymbol{a}_{\max}, -\boldsymbol{K}_a(\boldsymbol{q} - \boldsymbol{v}_{\max})\right), \boldsymbol{a}_{\min}\right),$$
$$\boldsymbol{a}_l = \min\left(\max\left(\boldsymbol{a}_{\min}, -\boldsymbol{K}_a(\boldsymbol{q} - \boldsymbol{v}_{\min})\right), \boldsymbol{a}_{\max}\right),$$

with the minimum and the maximum joint velocity limits $\boldsymbol{v}_{\min,\max}$ and the acceleration limits $\boldsymbol{a}_{\min,\max}$, $K_a > 0$ is a constant. The feasible acceleration region is illustrated in Figure 3. Analogous to the viability constraints, the feasible region of the acceleration is modified depending on the state of joint velocities. This technique effectively prevents overshooting.

## 2.4 Error Correction and Control Action Selection

For time-continuous systems, the state is obtained at a certain sampling rate and the action is applied for a certain period. This time discretization results in constraint violations at each time step. Therefore, we add an error correction term. We construct a P-controller with a diagonal matrix $\boldsymbol{K}_c$ for the constraints

$$\begin{bmatrix} \ddot{\boldsymbol{q}}_E \\ \dot{\boldsymbol{\mu}}_E \end{bmatrix} = -\boldsymbol{J}_c^\dagger \boldsymbol{K}_c \boldsymbol{c}(\boldsymbol{q}, \dot{\boldsymbol{q}}, \boldsymbol{\mu}). \tag{10}$$

Combining (9) with (10), we get the joint acceleration applied to the system

$$\begin{bmatrix} \ddot{\boldsymbol{q}} \\ \dot{\boldsymbol{\mu}} \end{bmatrix} = -\boldsymbol{J}_c^\dagger(\boldsymbol{q}, \boldsymbol{\mu}) \left[ \boldsymbol{K}_c \boldsymbol{c}(\boldsymbol{q}, \dot{\boldsymbol{q}}, \boldsymbol{\mu}) + \boldsymbol{\psi}(\boldsymbol{q}, \dot{\boldsymbol{q}}) \right] + \boldsymbol{N}_c(\boldsymbol{q}, \boldsymbol{\mu}) \boldsymbol{\alpha}. \tag{11}$$

The first term on the RHS is the necessary accelerations/velocities to maintain the constraints and the second term on the RHS is the tangent acceleration that can be explored freely. Figure 5 illustrates the vector field of error correction term and null space term of the circle constraint. The gray curves show the sampled trajectories converging to the constraint manifold due to the error correction.

The control action can be determined by $\boldsymbol{a} = \boldsymbol{\Lambda}(\ddot{\boldsymbol{q}})$ at different levels. For example, we can use the inverse dynamics model to calculate the joint torque when the robot is controlled via torque command. We can also apply the integration method to determine the desired positions/velocities, then use a sufficient accurate tracking controller (e.g., PID controller + Feedforward Term) to track the desired trajectory. However, the tracking errors could potentially cause hazardous constraint violations. In this paper, we control the joint torque calculated by a perfect dynamic model in simulation to simplify the analysis and to exclude the constraint violations caused by the tracking error of the controller. We present the block diagram of the controlling framework in Appendix A.

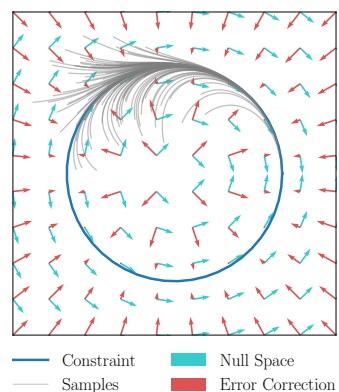

Figure 5: Vector Field of a Circle Constraint. The constraint $q_1^2 + q_2^2 - 1 = 0$ is the blue circle. The cyan arrows show the $\boldsymbol{N}_c \boldsymbol{\alpha}$ with $\boldsymbol{\alpha} = 1$. The red arrow demonstrate the error correction term $-\boldsymbol{J}_c^\dagger \boldsymbol{K}_c \boldsymbol{c}$. The gray lines show 100 trajectories from different initial points. All trajectories converge to the constraint manifold.

## 2.5 Null Space Convention

The orthogonal null space matrix $\boldsymbol{N}_c$ can be determined through SVD or QR decomposition. However, the representation of the null space bases is not unique. It is difficult to preserve the consistency

---

**Algorithm 1: ATACOM**

**Input:** Constraint: $\boldsymbol{f}, \boldsymbol{g}, \boldsymbol{J}_f, \boldsymbol{J}_g, \boldsymbol{b}_f, \boldsymbol{b}_g$. Scale parameter: $\boldsymbol{K}_c, \boldsymbol{K}_f, \boldsymbol{K}_g$. Time step $\Delta T$.

1 **for** *each episode* **do**
2     Initial feasible state $\boldsymbol{s}_0$, slack variable $\boldsymbol{\mu}_0$.
3     **for** *each time step $k$* **do**
4         Sample policy action $\boldsymbol{\alpha}_k \sim \pi(\cdot|\boldsymbol{s}_k)$.
5         Observe the $\boldsymbol{q}_k, \dot{\boldsymbol{q}}_k$ from $\boldsymbol{s}_k$.
6         Compute $\boldsymbol{J}_{c,k} = \boldsymbol{J}_c(\boldsymbol{q}_k, \boldsymbol{\mu}_k)$, $\boldsymbol{\psi}_k = \boldsymbol{\psi}(\boldsymbol{q}_k, \dot{\boldsymbol{q}}_k)$, $\boldsymbol{c}_k = \boldsymbol{c}(\boldsymbol{q}_k, \dot{\boldsymbol{q}}_k, \boldsymbol{\mu}_k)$.
7         Compute the RCEF of tangent space basis of $\boldsymbol{N}_c^R$
8         Compute the tangent space acceleration $[\ddot{\boldsymbol{q}}_k \; \dot{\boldsymbol{\mu}}_k]^\mathsf{T} \leftarrow -\boldsymbol{J}_{c,k}^\dagger [\boldsymbol{K}_c \boldsymbol{c}_k + \boldsymbol{\psi}_k] + \boldsymbol{N}_c^R \boldsymbol{\alpha}_k$
9         Clip the joint acceleration $\ddot{\boldsymbol{q}}_k \leftarrow \mathrm{clip}(\ddot{\boldsymbol{q}}_k, \boldsymbol{a}_l, \boldsymbol{a}_u)$
10         Integrate the slack variable $\boldsymbol{\mu}_{k+1} = \boldsymbol{\mu}_k + \dot{\boldsymbol{\mu}}_k \Delta T$
11         Apply the control action $\boldsymbol{a}_k = \boldsymbol{\Lambda}(\ddot{\boldsymbol{q}}_k)$ to the environment.
12         Observe the next state $\boldsymbol{s}_{k+1}$ and reward $r_k$ from the environment.
13         Provide the transition tuple $(\boldsymbol{s}_k, \boldsymbol{\alpha}_k, \boldsymbol{s}_{k+1}, r_k)$ to the RL algorithm

---

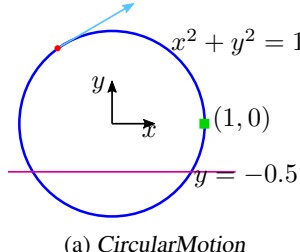

(a) *CircularMotion*

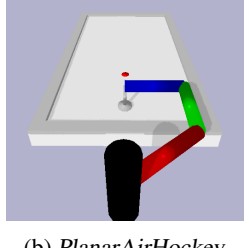

(b) *PlanarAirHockey*

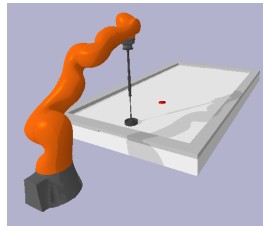

(c) *IiwaAirHockey*

Figure 6: Experiment Environments

of the null space bases computed by the numerical decomposition method [34, 35]. To solve this issue, we propose a convention to ensure the uniqueness of the null space bases.

Each column of the null space matrix $N_c$ is a unit vector indicating a direction of $[\ddot{q}\ \dot{\mu}]^\mathsf{T}$. However, this unit vector could sometimes contribute majorly to the part of the slack variable and the entries for the joint accelerations could be very small. As a result, the joint acceleration obtained from $\alpha \in [\alpha_{\min}, \alpha_{max}]$ can only cover a small region of the acceleration. As illustrated in Figure 4, the red arrow is a unit basis vector of the tangent space, and the reachable joint acceleration by a universal scaling factor could only cover part of the feasible joint acceleration, as the red area shown in Figure 4. To alleviate the previously mentioned issue, we compute the Reduced Column Echlon Form (RCEF) of the null space matrix $N_c^R = \text{RCEF}(N_c)$. Given that the RCEF of a matrix is unique, we obtain unique bases of the null space. In addition, for RCEF, each row containing a leading 1 has zeros in all its other entries. Generally speaking, there exist $N$ independent joints whose acceleration can be solely determined by $\alpha$, where $N$ is the dimensions of the null space. Also, we can define the feasible range of $\alpha$ as $\alpha_i \in [\ddot{q}_{i,\min}, \ddot{q}_{i,\max}]$. Through this convention, the joint acceleration is able to cover the full feasible range. Null space bases and feasible region are shown as the blue vector and the blue area in Figure 4.

## 3  Experiments and Evaluation

To illustrate the properties of our approach, we demonstrate three different experiments in this section. We first demonstrate a toy task, *CircularMotion*, shown in Figure 6a. In this task, we consider state equality, inequality, and velocity constraints. Secondly, we show a robotic environment with only inequality constraints, *PlanarAirHockey* shown in Figure 6b. A 3 DoF planar robot playing the hitting task in the air hockey scenario while keeping the end-effector inside the table boundary and the robot's joint positions and velocities within its limits. Finally, we demonstrate, *IiwaAirHockey* in Figure 6c, a 7-DoF KUKA IIWA robot learning the hitting task in the simulator. In addition to the constraints of the 3-dimensional task, we add an equality constraint to ensure that the robot end-effector stays on the table surface. More details can be found in the Appendix B and D.

***CircularMotion.***   In this task, shown in Figure 6a, the red point tries to move along a unit circle in 2D space while keeping the velocity of each direction below the velocity limits and maintaining the position above a certain height. The objective is to reach the target point (green square) located in $(1, 0)$. The control action is the acceleration $a = [\ddot{x}\ \ddot{y}]^\mathsf{T}$.

We compare ATACOM with two other approaches for the task. (i) *TerminatedCircularMotion* where the episode terminates when the maximum constraint violations up to a threshold. (ii) *ErrorCorrectionCircularMotion*, where the error correction term in (10) is added before the action is applied to the environment. We test five model-free RL algorithms (SAC, DDPG, TD3, TRPO, and PPO implemented in *Mushroom-RL* [36]) for each approach.

Figure 7 shows the learning curve and constraint violations of all test RL algorithms for ATACOM. Every algorithm is able to improve the learning performance and SAC outperforms the others methods, which matches our expectations. Figure 7b and 7c show the maximum constraint function and maximum joint velocity constraints at each time step. It can be shown that the maximum constraint violations during the whole learning process remain small. The velocity limit violations are zero after 30 epochs which means the learned policies try to fully exploit the constraints.

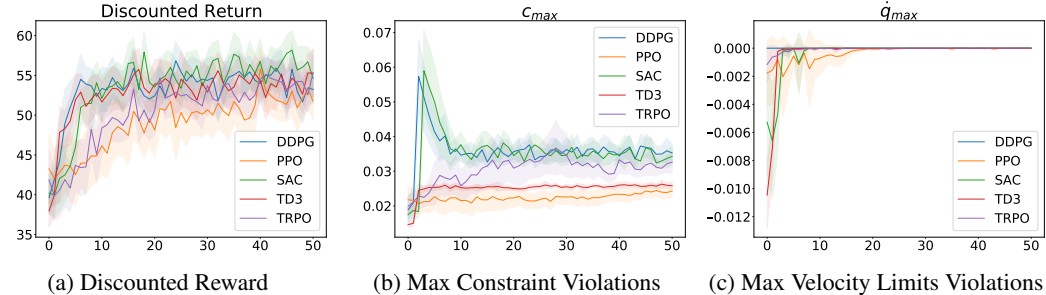

(a) Discounted Reward    (b) Max Constraint Violations    (c) Max Velocity Limits Violations

Figure 7: ATACOM for the *CircularMotion*. 7a shows the discounted return at each epochs. 7b and 7c shows the maximum constraint violations and maximum joint velocity limits violations.

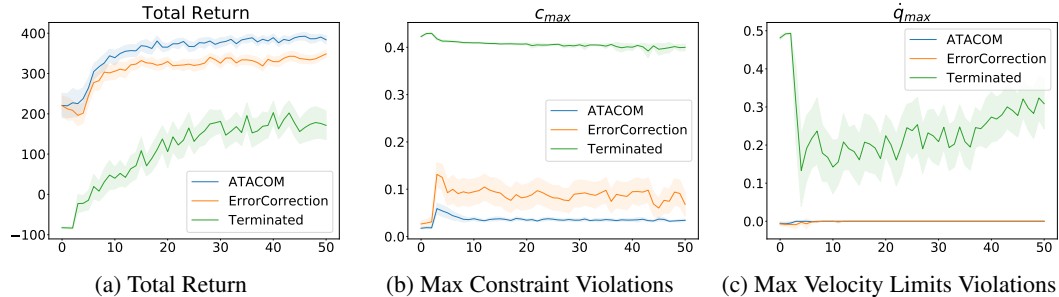

(a) Total Return    (b) Max Constraint Violations    (c) Max Velocity Limits Violations

Figure 8: Comparison between ATACOM, *Terminated-*, and *ErrorCorrectionCircularMotion*.

In Figure 8 we compare ATACOM, *TerminatedCircularMotion*, and *ErrorCorrectionCircularMotion*. We select the best learning algorithms for each approach (SAC for all cases). Compared to the baselines, our method focus on a lower-dimensional exploration space that avoid the constraint violations while the others do not. We can conclude from the Figures that ATACOM has not only lower constraint violations but also better learning performance among the three approaches.

**PlanarAirHockey.** In this experiment, we apply a 3-joints planar robot for the air-hockey hitting task, as illustrated in Figure 6b. The end-effector of the robot is kept on the table surface and the objective is to hit the puck to the opponent's goal. In this environment, we only consider inequality constraints, i.e., the robot end-effector should stay inside the table's region, and the joint positions and velocities should not exceed its limits. The control action is the joint torque obtained by the inverse dynamics model. In this experiment, we assume the dynamics model is perfectly known to eliminate the constraint violation due to the tracking error or the model mismatch.

In this task, we compared ATACOM with the *SafeLayer* method proposed by Dalal et al.,[10] and the *Unconstrained* air-hockey environments. Since the *SafeLayer* method at the beginning requires a free exploration process to learn the constraint function, we only compare the learning performance and the constraint violations after this process. For the unconstrained environment, the robot is completely free to explore, and the episodes only terminate when the maximum episode step is reached. In this experiment, we only compare the best DDPG result after the parameter sweep, as the available implementation of *SafeLayer* only supports DDPG. Additional experiment of *PlanarDefend* can be found in Appendix C.2

The result is shown in Figure 9. We can see that ATACOM have the best learning performance and the minimum constraint violations among the three methods. *SafeLayer* did not learn the constraint function of joint velocities properly. Furthermore, the learned constraints appear to be too restrictive to learn a good policy. Compared to the method of *Unconstrained* approach, although ATACOM has the same dimension as the *Unconstrained*, ATACOM explores only in the feasible region while the *Unconstrained* approach explores the whole state-action space. This consideration explains why ATACOM outperforms the baselines in terms of learning performances.

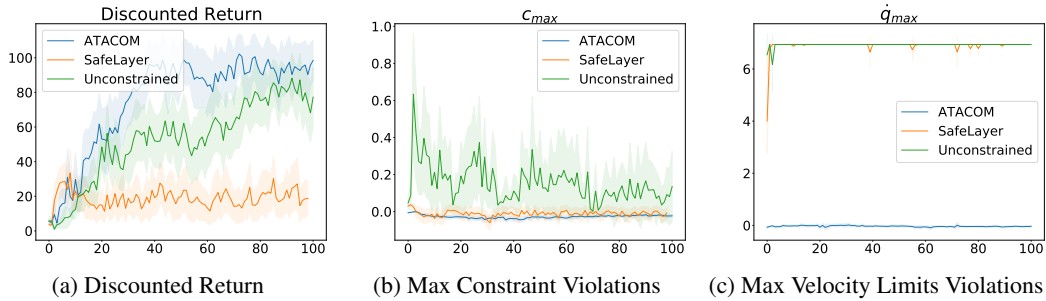

(a) Discounted Return     (b) Max Constraint Violations     (c) Max Velocity Limits Violations

Figure 9: Comparison between ATACOM, *SafeLayer* [10], *Unconstrained* AirHockey in DDPG.

***IiwaAirHockey.*** In the third experiment, we demonstrate the same air-hockey hitting task with a KUKA LBR IIWA14 Robot in the Pybullet simulator. In this task, we add an equality constraint to ensure the end-effector stays on the table surface. We also add inequality constraints to avoid collision of the end-effector and joint limit constraints as mention in the *PlanarAirHockey* task. In addition, we also add inequality constraints to avoid the collision between the 4/6-th link and the table. We enforce the joint velocity limits in the simulation as the real-world's KUKA controller does. We compare the impact of different simulation step sizes. The step size refers to the sampling frequency in the real world.

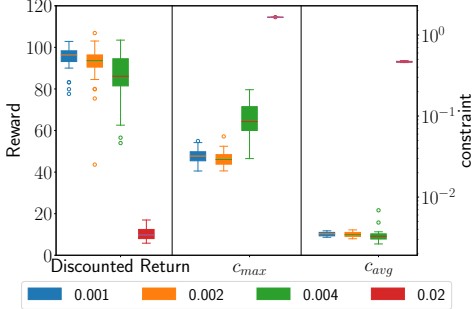

Figure 10: Box Plot of *IiwaAirHockey* with different choice of time step size.

At each simulation step, the torque is computed by the previous agent's action until the new control action is received. The error correction term is added at each time step. We choose the simulation step size as $0.02s$, $0.004s$, $0.002s$, and $0.001s$ and keep agent control frequency to be 50Hz.

Figure 10 demonstrates the discounted return at the final epoch, the maximum constraint violation $c_{max}$, and the average constraint violations $c_{avg}$ throughout the learning process. For a sufficiently small step size, such as $0.004s$, $0.002s$, $0.001s$, the learning agent is able to learn the hitting policy. When the step size is too big, e.g., $0.02s$, the error correction term dominates the control action and the agent has a poor learning performance. From the result in this experiment, we demonstrate that ATACOM is able to solve high dimensional tasks. The simulation result also provides us the guidance for the real-world application: The higher frequency of sampling and error correction, the smaller constraint violations will occur. In addition, we compare ATACOM with Riemannian Motion Policies [37] in Appendix C.4

## 4   Conclusion

In this article, we present ATACOM, a safe exploration method for Constrained RL based on the knowledge of the model and the mathematical formulations of constraints. ATACOM explores the tangent space of the constraint manifold. This exploration technique allows us to utilize any type of model-free RL method while maintaining the constraint violations below a small threshold. From the experiments, we have shown that ATACOM not only has small constraints violations but also better learning performance w.r.t. the other baselines. These performance gains occur because ATACOM only focuses on the safe region (from inequality constraint) and subspace (from equality constraint) of the whole state-action space.

However, our method still has some limitations. Our method requires a sufficiently accurate model or a good tracking controller. This assumption does not hold in most real-world applications since model errors, disturbances, and sensor noise could potentially cause unexpected constraint violations. To deploy this method in real-world robots, we will focus on the model mismatch problem and may require a backup policy to avoid too stringent constraint design. Furthermore, our current approach only focuses on the constraint with only controllable state $c(q) = 0$, even if preliminary results (Appendix E) suggests an extension into constraints with the uncontrollable state.

**Acknowledgments**

This project was supported by the CSTT fund from Huawei Tech R&D (UK). We acknowledge also the support provided by China Scholarship Council (No. 201908080039).

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
