# OpenReview forum: "Robot Reinforcement Learning on the Constraint Manifold"
_robot-learning.org/CoRL/2021/Conference — CoRL2021 Oral_

### Official Review · Reviewer_XePn · 2021-07-08

**Originality:** Good
**Technical Quality:** Good
**Clarity Of Presentation:** Good
**Impact:** 3

**Recommendation:**

Strong Accept: I recommend accepting the paper and will argue for my recommendation even if other reviewers hold a different opinion.

**Summary:**

This paper proposes to perform constrained RL by letting an RL agent choose robot velocities/accelerations directly on the constraint co/tangent manifold. These RL actions are mapped to robot join accelerations through standard inverse dynamics. To enforce inequality constraints, the proposed method introduces slack variables with associated dynamics that evolve according to the RL agent's actions. The method is demonstrated on a toy example and two arm air hockey tasks that require avoiding exceeding joint limits and an end-effector position constraint.

The authors have addressed all of my concerns and recommendations, and I think the paper is much stronger now. I already felt it was "accept" but feel more confident now!

(updated for final review)

**Issues:**

Note I have put my own references as [R#], listed in the REFERENCES section below; I denote references within the paper as [#]

Abstract:
- the first sentence could be more specific; WHICH practical issues are not normally considered in the ML literature?
- avoid contractions (e.g., it's)
- is the model of the agent uncertain? would be good to clarify in the abstract

1. Introduction
- The requirement that the constraint manifold is differentiable is quite strong, and would require smooth approximations or else subderivatives/Jacobians of non-smooth constraints (e.g., signed distance from a manipulator to a polytopic obstacle)
- it is good that the intro clearly states that the dynamics are perfectly known
- the idea of "no safe backup policy" is really odd since there are often feasible states from which a robot can never recover due to underactuation (e.g., a walking robot about to fall over)
- what prevents the proposed method from, e.g., accelerating a fully-actuated robot safely to a high speed such that the robot cannot avoid collision with an obstacle at the next time step?
- the authors should consider a couple additional methods in this space that are based on reachability analysis [R1,R2,R3]; each of these have advantages and limitations with respect to the proposed method
- a key issue with the Dalal et al. method in [10] is that it uses a linear approximation of the robot dynamics at the current state; it would be good to point out the proposed method's advantage of directly considering the nonlinear dynamics

2. Learning on the Constraint Manifold
- the transpose sign in (2) is $T$ instead of $\top$
- the discussion in section 2.5 seems unnecessarily detailed; I think you can just move the sentences on lines 187-188 and 199-200 to be under (11)
- I would imagine that, as the slack variables tend to zero, the RL agent tends to move more slowly, so that it doesn't violate the inequality constraints. But, I don't see a barrier function (e.g., $-log(\mu^2))$ to "encourage" this behavior, similar perhaps to [R4]. Maybe (6) creates exponential growth in the constraint-avoiding direction because $f(q) = -K_f\dot{f}(q,\dot{q})$?

3. Experiments and Evaluation
- it seems like an inverted pendulum formulation would be more widely-known than the CircularMotion task; I guess the point is to show how the proposed method stays on the equality constraint?
- the IiwaAirHockey task has equality constraints for the joint positions -- but not for the entire arm to not impact the table? this is frustrating because the really interesting constraint is the collision-avoidance constraint
- I think the two arm examples should really compare against the variety of methods in  controls and trajectory planning on manifolds, such as [R4,R5]

REFERENCES
[R1] Akametalu, A.K., Fisac, J.F., Gillula, J.H., Kaynama, S., Zeilinger, M.N. and Tomlin, C.J., 2014, December. Reachability-based safe learning with Gaussian processes. In 53rd IEEE Conference on Decision and Control (pp. 1424-1431). IEEE.

[R2] Shao, Y.S., Chen, C., Kousik, S. and Vasudevan, R., 2021. Reachability-based trajectory safeguard (rts): A safe and fast reinforcement learning safety layer for continuous control. IEEE Robotics and Automation Letters, 6(2), pp.3663-3670.

[R3] Krasowski, H., Wang, X. and Althoff, M., 2020, September. Safe reinforcement learning for autonomous lane changing using set-based prediction. In 2020 IEEE 23rd International Conference on Intelligent Transportation Systems (ITSC) (pp. 1-7). IEEE.

[R4] Bylard, A., Bonalli, R. and Pavone, M., 2021. Composable Geometric Motion Policies using Multi-Task Pullback Bundle Dynamical Systems. arXiv preprint arXiv:2101.01297.

[R5] Ratliff, N., Zucker, M., Bagnell, J.A. and Srinivasa, S., 2009, May. CHOMP: Gradient optimization techniques for efficient motion planning. In 2009 IEEE International Conference on Robotics and Automation (pp. 489-494). IEEE.

**Reviewer Expertise:**

Very good: Comprehensive knowledge of the area

**Strengths And Weaknesses:**

STRENGTHS
- The proposed idea is very clean and easy to understand, and the paper is mostly well-written
- The method outperforms an existing safe RL method
- The study of multiple RL agents and sampling times (simulation step sizes) is helpful for the reader to understand which heuristics matter for tuning the performance of this method
- I really like the idea of the slack variables evolving; it's clever to pull the slack variables "out" of an optimization loop and use them to represent constraint dynamics. This might be a standard technique in the optimization literature, but it's neat to see it applied to RL.

WEAKNESSES
- The method requires twice-differentiable constraints and a known robot model (which the paper admits)
- The method only considers learning instantaneous actions, as opposed to learning trajectories; so, it's unclear why the RL agent wouldn't drive a robot to an unrecoverable state from which constraint violation is inevitable
- In the experiments, there are no collision avoidance of the arm body with obstacles or self-collisions; but these are the really interesting and difficult constraints to handle, and it's unclear how to extend the method to handle them


**Summary Of Recommendation:**

I would like the authors to:
1. explain why the method wouldn't accelerate a robot to states from which recovery is impossible,
2. explain how the method might extend to uncertainty, underactuation, and non-smoothness in the dynamics and constraints,
3. compare against non-RL based techniques for constrained control and/or trajectory planning

Ideally the authors could also demonstrate the method on, e.g., a walking humanoid task or something really challenging to show how this method significantly outperforms any existing safe RL approaches (I don't know of any approaches that can learn to walk from scratch without falling/crashing).

---

> ### Author Response · Authors · 2021-08-24
> **Response**
>
> We thank the reviewer for the detailed analysis presented in the review. We will try to clarify all the concerns in the following response.
>
> > _"The method requires a known robot model."_
>
> We use the perfect model assumption to simplify the presentation and to provide a theoretically sound approach. In practice, it is also possible to use desired joint positions/velocities together with a tracking controller. This tracking error could potentially bring additional issues to the problem, like the interplay between the error in the constraints and tracking performance.
>
> > _"explain why the method wouldn't accelerate a robot to states from which recovery is impossible,"_
>
> > _"I would imagine that [...] because $f(q) = -K_f \dot{f}(q, \dot{q})$?"_
>
> To prevent this issue, we introduced the viability constraint to avoid getting stuck in an unrecoverable state. The discussion can be found in the response of Reviewer c13z.
>
> > _"In the experiments, there is no collision avoidance [...] unclear how to extend the method to tackle them. "_
>
> > _"the IiwaAirHockey task [...] is the collision-avoidance constraint"_
>
> In the PlanarAirHockey/IiwaAirHockey environment, we applied the constraint to prevent collision with the table edge. We apologize for not mentioning in the paper that we also added a constraint to prevent the collision between the elbow joint with the table in the IiwaAirHockey environment. You can find the missing details in the appendix (even in the first submission version). This obstacle avoidance constraint can be applied to every link. However, in the paper, we only consider static obstacles. In addition, it is also possible to extend our method with moving obstacles, discussions can be found in the response to Reviewer c13z.
>
> > _"explain how the method might extend to uncertainty, underactuation, and non-smoothness in the dynamics and constraints,"_
>
> For the stochastic dynamics and constraints, there are several directions that we would like to explore in the future, such as the expectation of the constraint, the chance constraint. Applying our method would be trivial if the stochastic constraint can be formulated as a differentiable function w.r.t controllable state.
>
> We did not focus on the problem of underactuation in this paper. One possible solution is to use more restrictive constraints to exclude states from the safety set where underactuation is a problem.
>
> For non-smooth dynamics such as contact, we think it is still possible to use the method as our method only requires smooth constraints and invertible dynamics. Obtaining a smooth function approximator is trivial: we can sample a sufficiently high amount of robot states in simulation, evaluate the non-smooth constraint functions and approximate them with a smooth relaxation constraint model. To avoid constraint violations, we can use conservative constraints (e.g., use a safe margin).
>
> > _“the idea of "no safe backup policy" [...] a walking robot about to fall over)”_
>
> In our model-based setting, a backup policy is not needed as we have the viability constraint (see the reply to Reviewer c13z). However, to enforce these constraints, the designer may be more restrictive. Using a backup policy would still be appropriate when these constraints are too restrictive or difficult to implement for particular applications, e.g., the standup policy for the humanoid environment.
>
> > _“Ideally the authors could also [...]  learn to walk from scratch without falling/crashing).”_
>
> We agree with the reviewer that the humanoid task is an important and interesting task for safe reinforcement learning. This task raises many issues that prevent the application of reinforcement learning to similar real-world platforms. However, developing a sufficient set of constraints to avoid falling is almost as hard as solving the original problem. Indeed, designing a set of constraints to maintain stability is the crucial point in the model-based locomotion controllers design. Nevertheless, our approach can be applied to some subtasks of the original problem, e.g., avoiding self-collision. This approach can be particularly beneficial in a hierarchical Reinforcement Learning setting.
>
> > _“compare against non-RL based techniques for constrained control and/or trajectory planning”_
>
> > _“I think the two arm examples [...] such as [R4,R5]”_
>
> Thanks for your suggestion. We are also working in this direction. We will compare the behavior in the real robot platform in the future. The mentioned method CHOMP [R5] would not satisfy the real-time constraint for the air hockey task. PBDS [R4] focuses on reactive planning and control could be potentially feasible on this problem. Unfortunately, the latter work is quite new, and we don’t have sufficient time to provide a comparison during this review process.
>
> Thanks for your advice on improving the clarity. We will try to revise them in the paper.

---

> > ### Comment · Reviewer_XePn · 2021-08-27
> > **Response to Authors' Comments**
> >
> > Thanks for the detailed response to myself and Reviewer c13z :)
> >
> > > In our model-based setting, a backup policy is not needed as we have the viability constraint (see the reply to Reviewer c13z). However, to enforce these constraints, the designer may be more restrictive.
> >
> > I think, as long as you make the restrictions of your method crystal clear in the paper, it will be fine. I'm already at an "accept" for this paper, and I feel the authors are addressing my concerns, at the very least as statements for future work.
> >
> > > The mentioned method CHOMP [R5] would not satisfy the real-time constraint for the air hockey task. PBDS [...] is quite new, and we don’t have sufficient time to provide a comparison during this review process.
> >
> > I realize that it is challenging to do a full comparison during the review process. The suggestion to compare against CHOMP is because it is well-known and has a good implementation available; the comparison would be to demonstrate path quality (e.g., if the same loss/cost function is used for CHOMP and ATACOM), as opposed to real-time performance. Also, saying that your method is faster than CHOMP, while guaranteeing safety, is a big win!
> >
> > It is also clear that [R5] (PBDS) is a very new work; however, maybe comparing against [Riemannian Motion Policies](https://arxiv.org/abs/1801.02854) would be more tractable?

---

> > > ### Author Response · Authors · 2021-08-29
> > > **Response to Reviewer's Suggestions**
> > >
> > > Thanks for the suggestions. We think that the suggested comparison with RMP could be very insightful, as this approaches encode a "soft" version of the constraint satisfaction problem. Also, this framework is closer to our approach w.r.t. planning methods. We will try to include this comparison in the final version.
> > >
> > > About the planning method, the experiment proposed could be indeed quite interesting, but it would need a careful definition and preparation in order to make a fair comparison between planning and learning methods and avoid wrong claims. Thus, we will perform these experiments in future works.

---

> > > > ### Comment · Reviewer_XePn · 2021-09-01
> > > > **thanks!**
> > > >
> > > > Great! Thanks for addressing all of my concerns. I hope the authors feel this review process has made their paper much stronger.

---

### Official Review · Reviewer_c13Z · 2021-07-24

**Originality:** Very Good
**Technical Quality:** Very Good
**Clarity Of Presentation:** Good
**Impact:** 4

**Recommendation:**

Weak Accept: I recommend accepting the paper, but will not argue for my recommendation if the majority of other reviewers have a different opinion.

**Summary:**

**Update:**

I appreciate the thorough response from the authors. The explanation and additional experiment address my main concern, about applicability to problems with constraints on uncontrollable parts of the state, so I have raised my score.

-----------

This paper proposes an approach, called ATACOM, that turns a constrained policy optimization problem into an unconstrained problem. The main idea is that rather than directly picking an action, the policy instead selects which direction to move in the tangent space of the constraint manifold (i.e., the states where the constraints are satisfied). This direction specifies the desired velocity, acceleration, or other higher-order derivative of the controllable state, and the action that achieves this is then executed. A key assumption of this approach is that the dynamics model is known.

This paper shows that the constraint values do not change, regardless of which tangent space direction is chosen by the policy—so if the constraints are satisfied at the initial controllable state q_0, then they will remain satisfied for the entire trajectory if ATACOM is used.

The empirical evaluation combines ATACOM with several state-of-the-art deep RL algorithms, and shows that it leads to minimal constraint violations, and outperforms existing approaches: early termination, error correction, and adding a safety layer.

**Issues:**

- What happens if the trajectory does not start at a feasible point? Will the agent be able to reduce the constraint violation, or is it unable to because the policy's choice of $\alpha$ doesn't change the constraint values?
- Why does the max violation never reach zero in the CircularMotion experiments?
- I would appreciate a more detailed explanation of why the null space basis can be viewed as the tangent space basis for the constraint manifold (lines 138-139). In general, the paper would be strengthened by adding more intuitive explanations throughout the method section.
- I recommend that the paper is edited to fix grammatical errors and typos.

**Reviewer Expertise:**

Good: General knowledge of the area

**Strengths And Weaknesses:**

**Strengths:**

- The approach is theoretically grounded.
- This approach can handle several realistic constraints for robot learning: providing a continuous velocity command, bounding the acceleration, and accounting for time discretization.
- The approach is flexible, and can be combined with any deep RL algorithm for learning the policy. The experiments support this.

**Weaknesses:**

- As is mentioned in the paper, the main limitation is that this approach requires that the dynamics model is perfectly known. This is typically an unrealistic assumption for real-world robotic settings.
- Another limitation is that the constraints can only apply to parts of the state that the robot can directly control, i.e. joint angles / velocities / accelerations. But there are many examples in which we may want to enforce constraints on parts of the state that the robot indirectly controls, for instance not bumping into dynamic obstacles while moving around, or not spilling water while carrying a cup. It is unclear how these could be expressed in terms of constraints on only controllable parts of the state.
- The description of the method needs more intuitive explanations, to make it accessible to those who are not already familiar with manifolds and tangent spaces. For instance, a more intuitive description of $\mathcal{A}$ is that $\mathcal{A}(\dot{q})$ produces the action $a$ that achieves velocity $\dot{q}$, based on my understanding.
- There are quite a few grammatical errors and typos.

**Summary Of Recommendation:**

This is a very interesting and promising idea, in the important research area of safe RL for robotics. It is both theoretically grounded and supported by empirical experiments. However, I have a couple of main concerns. I'm not convinced that setting constraints only on the controllable state (i.e., robot joint angles / velocities / accelerations) is enough to capture all/most of the kinds of safe robot RL problems. I don't think the paper addresses this limitation. I also found parts of the description of the method unclear and difficult to parse, so it's difficult to evaluate exactly what the method is doing.

---

> ### Author Response · Authors · 2021-08-24
> **Response to major concerns**
>
> We thank the reviewer for the constructive and insightful criticism. We will improve the paper to clarify the concerns that the reviewer raised.
>
> > _“As is mentioned in the paper, the main limitation [...] real-world robotic settings.”_
>
> Thank you for pointing this out. First, we would like to clarify that we need only the robot model rather than the dynamics of the whole environment. We believe it is reasonable to have a “sufficiently good” dynamical model for the robotic task. Many safe learning papers use this assumption. For example, in the navigation problem, velocity is usually selected as control action, the dynamics of motor and wheel is assumed to be known [1, 2],  model-based safe learning approaches [2, 3, 4, 5]. In addition, our method can exploit invertible learned dynamical models. We can also send desired position/velocity to the robot together with low-level tracking controllers. These technical details depend on the given application and are not discussed due to the page limitation and simplifying the method presentation. However, this is still a crucial issue for the broad applicability of the method. Thus, we will study the problem of model mismatch and stochasticity in future works.
>
> > _"Another limitation is that the constraints [...] on only controllable parts of the state.”_
>
> > _“This is a very interesting and promising idea [...] I don't think the paper addresses this limitation. "_
>
> Thanks for pointing out this concern. The safe RL for robotics is a very general topic that has different meanings from different perspectives. For example, the researchers from the RL perspective will focus on preventing the accumulative cost of the constraint above the threshold. Instead, researchers from a control perspective formulate the safe problem as preventing the robot from an unrecoverable state (such as pendulum). The focus of this paper is to propose a safe exploration method that focuses on local safety.
>
> Regarding the reviewer's concern, we can further divide the problem by two criteria, i.e., with/without uncontrollable states and known / unknown dynamics. 1). The problem without an uncontrollable state and with known dynamics is discussed in the paper. 2). As to the problem without any uncontrollable state and with unknown dynamics, we can integrate our method with a tracking controller (e.g., PD controller). 3). For the problem with the uncontrollable state and with a known model, it would also be possible to extend our method as follows: given the controllable $q$ and the uncontrollable state $p$, we can write the constraint manifold as $c(q, p) = 0$. The time derivative of the constraint is $\dot{c}(q, p) = J_q \dot{q} + J_p \dot{p}$. By setting $\dot{c}(q, p) = 0$. We get $\dot{q} = -J_q^{\dagger} (J_p \dot{p}) + N_q \alpha$. We can sample $\alpha$ while the constraints remain satisfied. This method requires the access of $\dot{p}$, which can be obtained by a state observer, e.g., Kalman Filter. We will add a simple experiment of dynamical collision avoidance for the uncontrollable state with known dynamics in the appendix. However, we need to study this problem more carefully studied in the future, e.g., viability constraints.  4). As to the problem of uncontrollable states with unknown dynamics, it is challenging to provide strong safety guarantees, as our method does. Indeed, in adversarial environments, such as prey-predator or even chess, there is no set of local constraints that can prevent you from reaching an unsafe state (being surrounded and captured for the prey-predator scenario, avoiding a checkmate in the second scenario). To ensure this safety, we should employ learning methods that will violate the learning process's constraints. The proposed framework allows us to easily incorporate other classes of safe learning agents since the learning agent selection is arbitrary. However, using our framework, we can enforce safety in a much stronger way on the constraint we support.
>
> Regarding the concern of the indirectly controllable state, there are no general solutions to our best knowledge. The solutions will be task-specific. For example, we can treat holding a cup while avoiding spilling as a fully controllable task if the robot is grasping the cup. For the task of pushing an object avoiding collision, the controllability will depend on the contact point.
>
> [1] Chow, Y. et al. Lyapunov-based Safe Policy Optimization for Continuous Control. *RL4RealLife Workshop in ICML 2019.*
>
> [2] Koller, T., et al. Learning-Based Model Predictive Control for Safe Exploration. *CDC 2018.*
>
> [3] Berkenkamp, F., et al. Safe Model-based Reinforcement Learning with Stability Guarantees. *NIPS 2017.*
>
> [4] Aswani, A., et al. Provably Safe and Robust Learning-based Model Predictive Control. *Automatica, 2013.*
>
> [5] Cowen-Rivers, A. I., et al. SAMBA: Safe Model-Based & Active Reinforcement Learning. *Arxiv.*

---

> ### Author Response · Authors · 2021-08-24
> **Response to minor concerns**
>
> Here is the response to the minor concerns for the reviewer:
>
> > _“The description of the method needs more intuitive explanations, to make it accessible to those who are not already familiar with manifolds and tangent spaces. For instance, a more intuitive description of A is that A(q) produces the action a that achieves velocity \dot{q}, based on my understanding.”_
>
> > _“There are quite a few grammatical errors and typos.”_
>
> > _“I would appreciate a more detailed explanation of why the null space basis can be viewed as the tangent space basis for the constraint manifold (lines 138-139). In general, the paper would be strengthened by adding more intuitive explanations throughout the method section.”_
>
> Thanks for the advice. We will revise the paper and add a short explanation and references about the manifolds and tangent space.  For the action selection part, we will explain using an illustrative example.
>
>
> > _“I also found parts of the description of the method unclear and difficult to parse, so it's difficult to evaluate exactly what the method is doing.”_
>
> To improve the paper's clarity, we will add a block diagram of the controller in the appendix and provide the link to the code in the final version.
>
> > _“What happens if the trajectory does not start at a feasible point? Will the agent be able to reduce the constraint violation, or is it unable to because the policy's choice of $\alpha$ doesn't change the constraint values?”_
>
> In this paper, we assume the trajectory starts from a feasible point or is sufficiently close to the constraint manifold. Starting from a feasible point or having an initial feasible policy is a general assumption in many references, e.g. [6, 7, 8].
>
> In addition, the viability constraint (section 2.2) will drive the system back to the safe region. As illustrated in Figure 3, when the constraint violates, i.e., $g(q)>0$. The maximum allowable velocity of the constraint function $\dot{g}$ is smaller than zero (blue area). The supremum of the constraints velocity implies that the dynamics of the constraint will move the constraint value back to the safe region. The idea of viability constraint is analogous to the Constraint Barrier Function (CBF) [9] and Safe Set Algorithms [10].  All these types of constraints address the property of forward invariance and finite-time convergence. We add a P controller to correct the constraint violations at each timestep, ensuring constraint satisfiability. We will show a simple example explaining this property in the paper.
>
> > _“Why does the max violation never reach zero in the CircularMotion experiments?”_
>
> Our method is discussed in the continuous form. But in practice, the algorithm is implemented in discrete time steps. At each time step, moving in the tangent direction of the manifold will result in small violations. We use a closed-loop error correction term to reduce the violations. In real-world applications, due to the existence of stochasticity, measurement error, and disturbances, it is reasonable to accept a small tolerance.
>
> [6] Chow, Y., et al. A Lyapunov-based Approach to Safe Reinforcement Learning. *NIPS 2018*.
>
> [7] Garcia, J., & Fernandez, F. Safe Exploration of State and Action Spaces in Reinforcement Learning. *Journal of Artificial Intelligence Research, 2012.*
>
> [8] Achiam, J., et al. Constrained Policy Optimization. *ICML 2017.*
>
> [9] Ames, A. D., et al.  Control Barrier Function Based Quadratic Programs for Safety Critical Systems. *CDC, 2014.*
>
> [10] Liu, C., & Tomizuka, M. (2014). Control in a safe set: Addressing safety in human-robot interactions. *Dynamic Systems and Control Conference (DSCC) 2014.*

---

### Official Review · Reviewer_n6jm · 2021-07-24

**Originality:** Very Good
**Technical Quality:** Very Good
**Clarity Of Presentation:** Very Good
**Impact:** 4

**Recommendation:**

Strong Accept: I recommend accepting the paper and will argue for my recommendation even if other reviewers hold a different opinion.

**Summary:**

This paper proposes a novel approach to efficiently learn robotics tasks in simulation while satisfying safety constraints by leveraging the knowledge of the robot’s model and mathematical definition of the constraints. The proposed method (ATACOM) explores in the tangent space of the constraint manifold by including the model and constraints information into the computation of the action. This allows to convert the constrained RL problem into an unconstrained RL problem, which allows the application of any type of model-free RL algorithm to solve the task. In addition, the proposed method does not require an initial feasible policy or a safe backup policy as other safe exploration methods found in the literature. The effectiveness of the method is shown in three different experiments with different levels of difficulty and both equality and inequality constraints. The results show ATACOM has comparable or better performance as other safe exploration methods while keeping the constraints below an acceptable tolerance.

**Issues:**

In the abstracts the authors mention that although many safe exploration and constrained reinforcement learning algorithms exist in the ML literature, these are not applicable to real robotics tasks, which is a misleading statement. There are several works about Safe Exploration in RL that have been applied in simulated robotic tasks e.g. Berkenkamp et. all - “Safe Model-based Reinforcement Learning with Stability Guarantees” and even real robots e.g. Taylor et all - “Learning for Safety-Critical Control with Control Barrier Functions”.

Minor observations in the grammar:
“A variety of different practical considerations” is redundant.
“The ATACOM method can be summarized as followings:”
“When the starting from the point”


**Reviewer Expertise:**

Very good: Comprehensive knowledge of the area

**Strengths And Weaknesses:**

Strengths
- Although the idea of using model-based information to improve the learning is not new and the outcome of improving the efficiency of the method is expected, using the mathematical definition of the constraints to formulate the problem such that the exploration is done in the tangent space of the constrained manifold is a novel and interesting idea.
- The paper is clear and well written, and the authors provided a detailed explanation of their method with extensive evaluation and discussion of the results. I liked the authors included detailed information about the environments used for the experiments and the details about the implementation of their method in each of these environments e.g. actual rewards terms, equality and inequality constraints, parameters of the learning algorithm, structure of the NN, etc.
- The paper provides extensive testing and comparison with different methods and RL algorithms.

Weaknesses
- The downside of the method is its scalability to high-order complex systems since it requires knowing the dynamic model of the system. In addition, the mismatch between the simulation model and the real world model can present significant challenges for the implementation of the method in real world applications.


**Summary Of Recommendation:**

- My only recommendations for the authors would be to specify in detail the integration of the action space of the learning algorithm with the control action of the system. For example, what is the output of the NN and how it is integrated into the action/control input used for commanding the system.
- I think the comparison in Figure 9 does not really provide any insightful result. It is expected that lower sampling frequency has a negative impact in the performance of the method.
- It would be interesting to see a discussion about the use of different selections for the action space. For example, could the action space of the policy be the desired joint position and apply low-level PD controllers to track such desired positions?

---

### Meta-Review · Area_Chair_7BjM · 2021-08-14

**Recommendation:** Accept (Oral)
**Confidence:** 5

**Metareview:**

Summary: This paper presents a method for safe learning called ATACOM. The method transforms a constrained RL problem into an unconstrained problem. It is general for any model-free RL algorithm, accommodates both equality and inequality constraints, and is shown to be more effective than other baseline methods in simulation experiments.

Clarity: In general, the paper is clear in its motivation, overarching goal behind the proposed solution, and the analysis of its experimental results. That being said, the authors are encouraged to consider reviewer c13Z’s feedback in adding more information about the intuition behind constraint manifolds.

Quality: Reviewers appreciated the comparison of ATACOM with a breadth of other state-of-the-art methods and seem mostly convinced that the proposed method outperforms.

Originality: Reviewers were generally positive about the originality of the theoretical aspects of the proposed approach.

Significance: On the practical side, reviewers had mixed feedback about the usability of the proposed approach. There is a concern that assumptions are too restrictive and renders the algorithm not practically usable for real-world robots. (See “opportunities for improvement”).

Pros:
The paper presents a novel approach to safe learning, an area of strong interest to both robotics and ML communities.

Main opportunities for improvement:
I would suggest clearly characterizing and discussing the usability of the proposed approach to be a focus areas for the rebuttal and revision. All reviewers expressed concern that different assumptions in the algorithm (e.g. dynamics model must be known, constraints must be in terms of controllable parts of the state, smooth constraints, etc.) prevent actual application to real robots as claimed. Please see each reviewer’s individual review for specific details.

Aside from this, reviewers made more minor suggestions for improving the readability of the paper that should be addressed.

Thank you for considering our feedback, and we look forward to seeing the updated paper.

========== Final Decision

All reviewers recommend accept, with 2/3 recommending strong accept. The paper also received strong scores in all sub-categories from all reviewers. Most notably, 2/3 reviewers both gave it a 4 in impact and very good in originality. As a result, I think it would be appropriate to highlight this paper in the oral session.

---

> ### Author Response · Authors · 2021-08-30
> **Summary of the Revision**
>
> Dear Area Chair,
>
> Thanks for your time and suggestions. This is the summary of the modifications to the paper, addressing reviewers' concerns:
> 1. We add a block diagram in the appendix to improve the clarity of the method. In addition, we add some explanation about the action selection according to the suggestion of Reviewer c13Z and n6jm.
> 2. We clarify the assumption of the perfect model in the reply to the reviewer and the paper. This assumption is used in our experimental analysis to exclude the constraint violations due to the model mismatch or the trajectory tracking error. However, in practice, we could use a sufficiently good invertible robot dynamics model or an accurate tracking controller.
> 3. As Reviewer n6jm suggested, we simplified the experiment analysis of the sampling frequency. Instead, we demonstrate an illustrative example of the tangent space, error correction, and random trajectories starting from different points, shown in Figure 5.
> 4. We add a discussion about extending our method with uncontrollable states in Appendix E. We also add an experiment of collision avoidance task with four moving objects. The learning curve of SAC is shown in Appendix E.1. We also add a video demonstrating the learning behavior in the attached video.
> 5. We improve the clarity of constraint manifold, nullspace, and tangent space in the paper and add some related references.
> 6. We fixed the misleading claim in the abstract.
> 7. We fixed the typos, grammar errors, and missing details in the paper.

---

### Decision · Program_Chairs · 2021-09-13

**Decision:**

Accept (Oral)

**Comment:**

Summary: This paper presents a method for safe learning called ATACOM. The method transforms a constrained RL problem into an unconstrained problem. It is general for any model-free RL algorithm, accommodates both equality and inequality constraints, and is shown to be more effective than other baseline methods in simulation experiments.

Clarity: In general, the paper is clear in its motivation, overarching goal behind the proposed solution, and the analysis of its experimental results. That being said, the authors are encouraged to consider reviewer c13Z’s feedback in adding more information about the intuition behind constraint manifolds.

Quality: Reviewers appreciated the comparison of ATACOM with a breadth of other state-of-the-art methods and seem mostly convinced that the proposed method outperforms.

Originality: Reviewers were generally positive about the originality of the theoretical aspects of the proposed approach.

Significance: On the practical side, reviewers had mixed feedback about the usability of the proposed approach. There is a concern that assumptions are too restrictive and renders the algorithm not practically usable for real-world robots. (See “opportunities for improvement”).

Pros:
The paper presents a novel approach to safe learning, an area of strong interest to both robotics and ML communities.

Main opportunities for improvement:
I would suggest clearly characterizing and discussing the usability of the proposed approach to be a focus areas for the rebuttal and revision. All reviewers expressed concern that different assumptions in the algorithm (e.g. dynamics model must be known, constraints must be in terms of controllable parts of the state, smooth constraints, etc.) prevent actual application to real robots as claimed. Please see each reviewer’s individual review for specific details.

Aside from this, reviewers made more minor suggestions for improving the readability of the paper that should be addressed.

Thank you for considering our feedback, and we look forward to seeing the updated paper.

========== Final Decision

All reviewers recommend accept, with 2/3 recommending strong accept. The paper also received strong scores in all sub-categories from all reviewers. Most notably, 2/3 reviewers both gave it a 4 in impact and very good in originality. As a result, I think it would be appropriate to highlight this paper in the oral session.